# Bioactive Compounds and Pharmacological Properties of the Polypore *Fomes fomentarius*, a Medicinal Wild Mushroom Collected from Morocco

**DOI:** 10.3390/ijms26189215

**Published:** 2025-09-21

**Authors:** El Hadi Erbiai, Safae Maouni, Luís Pinto da Silva, Rabah Saidi, Zouhaire Lamrani, Joaquim C. G. Esteves da Silva, Abdelfettah Maouni, Eugénia Pinto

**Affiliations:** 1Centre for the Research and Technology of Agroenvironmental and Biological Sciences, CITAB, Inov4Agro, Universidadede Trás-os-Montes e Alto Douro, UTAD, Quinta de Prados, 5000-801 Vila Real, Portugal; 2Biology, Environment, and Sustainable Development Laboratory, École Normale Supérieure de Tétouan (ENS), Abdelmalek Essaadi University, Tetouan 93000, Morocco; maouni.safae88@gmail.com (S.M.); r.saidi@uae.ac.ma (R.S.); zh.amrani@yahoo.fr (Z.L.); amaouni@uae.ac.ma (A.M.); 3Chemistry Research Unit (CIQUP), Institute of Molecular Sciences (IMS), Department of Geosciences, Environment and Spatial Plannings, Faculty of Sciences, University of Porto, Rua do Campo Alegre s/n, 4169-007 Porto, Portugal; luis.silva@fc.up.pt (L.P.d.S.); jcsilva@fc.up.pt (J.C.G.E.d.S.); 4Department of Dermatology, Mohamed 5 University, Rabat B.P. 6527, Morocco; 5Laboratory of Microbiology, Biological Sciences Department, Faculty of Pharmacy, University of Porto, 4050-313 Porto, Portugal; 6Interdisciplinary Centre of Marine and Environmental Research (CIIMAR), University of Porto, 4450-208 Matosinhos, Portugal

**Keywords:** Moroccan mushroom, polypore mushrooms, *Fomes fomentarius*, bioactive compounds, pharmacological properties, volatile and non-volatile molecules

## Abstract

Polypore mushrooms have been widely recognized for centuries for their use in food and medicine due to their strong capacity to produce numerous biomolecules with beneficial effects on human health. *Fomes fomentarius* is one such species that remains poorly explored, particularly when growing in Morocco. Herein, this study aimed to characterize the bioactive compounds of *F. fomentarius* and evaluate its pharmacological properties. Spectrophotometric analysis showed that *F. fomentarius* revealed high levels of total phenolics (75.83 mg GAE/g dme) and flavonoids (37.62 mg CE/g dme). Gas chromatography–mass spectrometry (GC–MS) analysis identified 109 volatile and non-volatile compounds, primarily sugars (24), fatty acids (23), alcohols (10), organic acids (9), and terpenoids (6). In addition, liquid chromatography-mass spectrometry (LC-MS) analysis allowed the identification of 24 phenolic compounds, with isorhamnetin (2734.00 µg/g), *p*-hydroxybenzoic acid (409.00 µg/g), and kaempferol (351.10 µg/g) as the most abundant. Regarding pharmacological properties, *F. fomentarius* extract demonstrated strong antioxidant activity, with the DPPH radical-scavenging assay showing the highest potency, followed by *β*-carotene bleaching inhibition and ferric ion-reducing power, with EC_50_ (half maximal effective concentration) values of 114.40, 174.50, and 250.70 µg/mL, respectively. Additionally, it exhibited broad-spectrum antimicrobial activity against all seven human pathogenic microorganisms, with *Epidermophyton floccosum* being the most susceptible ((minimum inhibitory concentration (MIC)) = 2 mg/mL and minimal fungicidal concentration (MFC) = 4 mg/mL) and *A. fumigatus* the most resistant (MIC = 26.67 mg/mL and MFC ≥ 64 mg/mL). Overall, the result indicated that Moroccan *F. fomentarius* is a rich source of diverse bioactive compounds with potent antioxidant and antimicrobial activities, supporting its potential for various applications.

## 1. Introduction

Polypore mushrooms are a diverse group of *Basidiomycota* fungi, many of which are inedible because of their woody texture, although some species are consumed as food or used in traditional medicine. These fungi are characterized by their large fruiting bodies and widespread global distribution [1,2]. They have been utilized for centuries, particularly in Europe and Asia, where they hold cultural significance in Chinese, Japanese, and Korean medicinal practices [1,2,3,4]. Polypore fungi are renowned for their ability to produce a wide array of biologically active molecules, including terpenes, polysaccharides, lipids, proteins, phenolic compounds, esters, organic acids, alkaloids, nucleosides, and various other bioactive compounds. These bioactive molecules exhibit numerous health-promoting properties, including antioxidant, antimicrobial, anti-inflammatory, anticancer, antiviral, antidiabetic, anti-atherosclerotic, immunomodulatory, and neuroprotective effects [1,3,5,6,7,8]. Several polypore species are particularly well known for their medicinal properties, including *Ganoderma lucidum*, *G. applanatum*, *Inonotus cuticularis*, *I. hispidus*, *I. obliquus*, *Laetiporus sulphureus*, *Grifola umbellata*, *Meripilus giganteus*, *Piptoporus betulinus*, *Trametes hirsuta*, *T. versicolor*, *Wolfiporia cocos*, and *Fomes fomentarius* [1,3,5,6,7,8]. Despite their long-standing use, polypore mushrooms remain a promising and sustainable source for the discovery and development of novel pharmaceuticals and nutritional supplements beneficial to humans. For that, *F. fomentarius* was selected for further investigation due to its potential therapeutic properties.

*F. fomentarius* is a medicinal polypore mushroom, commonly known as the tinder fungus, belonging to the family Polyporaceae. It is a parasitic-saprophytic fungi playing a crucial role in forest decomposition and nutrient cycling [3,9,10]. This species has been recorded on multiple tree hosts across Europe, Asia, North America [11], and in diverse Moroccan ecosystems [12,13,14,15,16,17]. While technically edible, it is unpalatable and rarely consumed as food; however, its nutritional and functional properties make it a valuable source of bioactive compounds [11]. Traditionally, *F. fomentarius* has been employed in the treatment of various ailments, including inflammation, hemorrhage control, wound healing, immune enhancement, digestive improvement, oral ulcers, gastric and uterine disorders, gastroenteric diseases, hepatocirrhosis, and certain cancers [3,10,12,13,14,15].

Previous studies on *F. fomentarius* have reported the presence of numerous bioactive constituents, including phenolic compounds, sterols, terpenoids, polysaccharides, proteins, esters, alcohols, organic acids, ketones, and other volatile compounds [3,12,13]. Extracts and isolated bioactive compounds from this species have been associated with diverse biological activities, such as antioxidant, antimicrobial, antimutagenic, anti-inflammatory, antiviral, enzymatic modulatory, antiproliferative, antinociceptive, and antitumor effects [10,12,13,14]. Despite these promising properties, *F. fomentarius* remains poorly explored. Many of its volatile and non-volatile constituents have yet to be fully identified and quantified, and their biological activities, particularly potential anti-dermatophytic effects, require further investigation. Moreover, no studies to date have addressed the chemical characterization and biological activities of *F. fomentarius* populations growing abundantly in different regions of Morocco.

The present study aimed, firstly, to characterize the chemical compound profile of *F. fomentarius* by (i) determining the content of key bioactive compounds, including total phenolic compounds, total ascorbic acid, and total carotenoids, using spectrophotometric methods; (ii) identifying and quantifying individual polyphenols through liquid chromatography-mass spectrometry (LC–MS); and (iii) analyzing volatile and non-volatile molecules using gas chromatography-mass spectrometry (GC–MS). The second objective was to evaluate the antioxidant and antimicrobial properties of the methanolic extract of *F. fomentarius*. The results demonstrated that *F. fomentarius* is a rich source of various biologically active compounds with potent antioxidant and antimicrobial activities, highlighting its potential for various pharmaceutical and nutraceutical applications.

## 2. Results and Discussion

### 2.1. Extraction Yield and Bioactive Compounds Contents in F. fomentarius

The extraction of bioactive compounds from the polypore mushroom *F. fomentarius* harvested in Morocco was performed using methanol, yielding 7.59%, which is higher than previously reported methanolic extraction yields of 4.3% [16] and ethanolic extraction yields of 2.6% [12,16]. However, this yield was lower than that obtained using hot water extraction, which reached 10.4% [12].

As shown in Table 1, the dried fruiting bodies of *F. fomentarius* growing on *Acacia* species in Morocco contained a solid bioactive compound content predominantly composed of total phenolics (TPC), measured at 75.83 mg gallic acid equivalents (GAE) per gram of dry methanolic extract (dme). This was followed by total flavonoids (TFC), at 37.62 mg (+)-catechin equivalents (CE)/g dme, and ascorbic acid (AAC), at 3.43 mg L-ascorbic acid equivalents (AAE) per gram of dry weight (dw). Lower concentrations were observed for total tannins (TTC) and total carotenoids (*β*-carotene (*β*CC) and lycopene (LC)). These natural molecules are well-recognized antioxidants and contribute to a range of other biological activities.

The TPC found in Moroccan *F. fomentarius* (75.83 mg GAE/g dme) was notably higher than previously reported values from Turkey (8.58 mg GAE/g dme) [17], Serbia (26.41 mg GAE/g dme [13] and 43.06 mg GAE/g dry ethanolic extract, dee) [16], Tunisia (49.6 mg GAE/g, dee) [18], and Poland (53.13 mg GAE/g dee) [19]. Conversely, the Moroccan TPC was lower than those reported in a methanolic extract from Serbia [16], an ethanolic extract from South Korea [12], and a water extract from Spain [20], which had values of 82.54, 119.9, and 119.64 mg GAE/g of dried extracts, respectively.

Regarding TFC, *F. fomentarius* from Morocco contained 37.62 mg CE/g dme, which was comparable to the methanolic extract from the USA (38.13 mg propylgallate acid equivalent/g) and higher than the ethanolic extract from the same study (32.13 mg/g) [21]. Studies from Tunisia [18], Turkey [17], and Serbia [16] reported significantly lower flavonoid levels in their *F. fomentarius* samples, ranging between 1.2 and 11.47 mg/g of dried extract.

The TTC in the Moroccan mushroom was determined as 0.95 ± 0.07 mg CE/g dry weight (dw), which was substantially higher than the value found in the Tunisian mushroom (0.005 mg/g dw) [18].

Importantly, this study represents the first report on the content of other key bioactive compounds in *F. fomentarius*, including ascorbic acid (3.43 mg AAE/g dw), *β*-carotene (0.59 mg/g dme), and lycopene (0.19 mg/g dme).

### 2.2. Biomolecules Profiles of F. fomentarius by GC-MS

GC–MS, a powerful technique for biomolecule identification, was employed to characterize the biomolecules profile of the *F. fomentarius* methanolic extract. To detect both volatile and non-volatile molecules, the extract was processed into two sample types: diluted in chloroform for volatile compounds and derivatized with BSTFA (N,O-bis(trimethylsilyl)trifluoroacetamide) reagent in chloroform for non-volatile compounds. GC–MS analysis of the two resulting samples revealed 109 volatile and non-volatile compounds, representing over ten major biomolecular classes (Table 2 and Appendix A). A higher number of biomolecules was identified in derivatized extracts (70), while 39 constituents were detected in the extract diluted in chloroform (Table 2). The majority of identified biomolecules belong to the following groups: sugars (24), fatty acids (23), alcohols (10), organic acids (9), and terpenoids (6). Generally, alcohols, fatty acids, organic acids, phenols, and steroids were detected in both sample types, whereas alkaloids and alkanes were found exclusively in the chloroform-diluted sample, and amino acids and sugars were detected only in the derivatized sample.

In the methanolic extract diluted in chloroform, GC–MS analysis identified 39 volatile components, primarily belonging to the classes of phenols (40.30%), fatty acids (15.80%), and alkaloids (11.55%). The predominant compounds were 4-(3,5-di-tert-butyl-4-hydroxyphenyl)butyl acrylate (21.71%), 2,6-di-tert-butyl-4-(dimethylaminomethyl)phenol (18.16%), 6-ethoxy-2,2,4-trimethyl-1,2,3,4-tetrahydroquinoline (12.11%), botulin (10.00%), 5-benzylquinoline (9.87%), and linoelaidic acid (5.28%) (Appendix A; Table 2 and Appendix A). Phenolic compounds were the dominant group in the chloroform-prepared sample, and these are well-reported for their diverse pharmacological and biological activities [22,23,24,25]. Quinolines, present in high proportions in this sample, are known to exhibit diverse biological activities, including antimalarial, anti-inflammatory, anthelmintic, antibacterial, antifungal, anticonvulsant, and analgesic effects [26]. The triterpenoid compound betulin, identified among the primary constituents in the chloroform extract, has previously been reported in *F. fomentarius* and is associated with anticancer and anti-inflammatory properties [27,28,29,30]. The fatty acid group contained a relatively high number of biomolecules (Table 2), with linoelaidic acid (5.28%) and cis-vaccenic acid (4.21%) as the predominant compounds (Appendix A). Linoelaidic acid has been reported to possess strong anticancer activity [31]. In comparison, Prasad et al. [32] found that the volatile fraction of Indian *F. fomentarius* was dominated by i-phellandrene and α-phellandrene in hexane and ethyl acetate extracts, respectively.

In the derivatized extract, the GC–MS chromatogram revealed numerous peaks (Figure 1), corresponding to seventy volatile and non-volatile compounds in *F. fomentarius*, classified into six major groups: sugars (64.63%), fatty acids (14.72%), organic acids (5.36%), steroids (3.75%), alcohols (3.27%), and amino acids (1.08%), along with fourteen compounds from other groups (7.17%) (Table 2 and Appendix A). The predominant biologically active molecules were D-glucopyranose (14.64%), xylitol (11.62%), D-mannitol (9.75%), D-trehalose (8.07%), linoleic acid (3.81%), and elaidic acid (3.35%). Moroccan *F. fomentarius* was notably rich and diverse in sugars, comprising 24 identified molecules, including 19 monosaccharides (50.95%) and five disaccharides (13.68%) (Appendix A). This is the first study to report the sugar composition of *F. fomentarius* alcoholic extract. Among the monosaccharides, D-glucopyranose, xylitol, and D-mannitol were the most abundant, whereas D-trehalose (8.07%), 3-α-Mannobiose (2.46%), and maltitol (2.09%) were the predominant disaccharides. The identified sugars may possess various biological activities; notably, sugar alcohols have been reported to exhibit antioxidant, anticaries, antimicrobial, and anti-inflammatory effects [4]. The fatty acid fraction (Appendix A) comprised 11 molecules, with linoleic acid (3.81%), elaidic acid (3.35%), 4-oxohexanoic acid (2.64%), and palmitic acid (1.98%) as the predominant metabolites. Fatty acids, including these major compounds, are known for antioxidant, antimicrobial, antifungal, antibacterial, and tumor-promoting activities [33,34,35,36]. In a comparative study, Kalitukha and Sari [20] quantified 21 fatty acids in *F. fomentarius* from Spain, with erucic, palmitic, and stearic acids as the most abundant. Among the seven organic acids detected (Appendix A), malic acid (1.87%) was predominant. The steroids group (Appendix A) contained three molecules, primarily ergosterol (2.27%) and ergosta-7,22-dien-3β-ol (1.43%), previously reported in *F. fomentarius* from China [27,37] and Russia [29] and associated with antitumor activity [27,37]. Ergosterol is a key sterol with reported antioxidant, anti-inflammatory, antimicrobial, and anticancer properties, as well as a role in preventing common diseases [4]. Additionally, the derivatized methanolic extract of Moroccan *F. fomentarius* was rich in other bioactive constituents, including alcohols, amino acids, and nucleosides (Appendix A). Overall, GC–MS analysis of the two methanolic extract samples demonstrated that *F. fomentarius* growing on *Acacia* species in Morocco is rich in a diverse array of volatile and non-volatile compounds with broad nutritional and biologically beneficial properties.

### 2.3. Individual Polyphenols of F. fomentarius by LC–MS Analysis

Phenolic compounds are among the principal bioactive compounds in mushrooms, and LC–MS is a powerful technique for their characterization. In this study, LC–MS analysis of the hydro-methanolic extract of *F. fomentarius* revealed multiple individual polyphenols, visualized as peaks in Figure 2. Several of these were identified and quantified using 25 authentic commercial standards, along with their corresponding mass spectra, UV profiles, and retention times (Table 3). The predominant quantified compounds in the fruiting body were isorhamnetin (2734.00 µg/g dw), *p*-hydroxybenzoic acid (409.00 µg/g), kaempferol (351.10 µg/g), apigenin (295.10 µg/g), and chlorogenic acid (150.80 µg/g). In contrast, vanillic acid, naringin, and ferulic acid were detected in smaller amounts (2.19, 2.51, and 3.34 µg/g, respectively), while syringic acid was not detected (Table 3). Moreover, *F. fomentarius* contained a substantially higher concentration of flavonoids (3787.80 µg/g) than phenolic acids and related compounds (803.00 µg/g).

To date, only a few studies have characterized the individual polyphenols of *F. fomentarius*: one from Poland [19], one from Serbia [16], three from Turkey [38,39,40], and one from the Balkan region (Croatia, Serbia, and Bosnia and Herzegovina) [41]. Most studies reported ten or fewer compounds, primarily phenolic acids, whereas the Balkan study identified a broader range of phenolic and derived compounds. This contrasts with our findings, which revealed 24 compounds, including 14 phenolic acids and 10 flavonoids. The Polish [19] and Serbian [16] studies identified only phenolic acids, with *p*-hydroxybenzoic acid as the predominant compound, consistent with our phenolic acids profile. In Turkish *F. fomentarius*, Doğan et al. [38] and Dundar et al. [40] reported benzoic acid (175.13 µg/g) and *o*-coumaric acid (361.76 µg/g), respectively, as the major phenolic acids, compounds not analyzed in the present work. Both studies also detected syringic acid, which was absent in our sample as well as in the Polish and Serbian reports. Dundar et al. [40] found quercetin (486.46 µg/g) as the predominant polyphenol, a level substantially higher than that observed in Moroccan samples (99.47 µg/g). The results for *F. fomentarius* from the Balkan region revealed quinic acid and scopoletin as the predominant compounds, which were not detected in the Moroccan sample [41]. Furthermore, several phenolic compounds detected in the present study, including apigenin, apigenin 7-glucoside, ellagic acid, isorhamnetin, kaempferol, luteolin, luteolin 7-glucoside, methylparaben, naringin, rosmarinic acid, and vanillin, have not been previously reported in any studies of the *F. fomentarius* fruiting body. Overall, the high quantity and diversity of phenolic compounds identified in the *F. fomentarius* fruiting body underscore its potential as a rich source of antioxidants. These compounds are well-known for their wide range of biological activities, including anti-inflammatory, antimicrobial, anti-tyrosinase, anti-osteoporotic, and anticancer effects, and they have applications in the nutraceutical and cosmetic industries [22,23,24,25].

### 2.4. Antioxidant Properties of F. fomentarius

Antioxidants are a large group of beneficial molecules known to mitigate numerous chronic diseases such as inflammation, aging, cancer, cardiovascular diseases, and anemia [42]. Natural antioxidants are increasingly preferred for human health applications and have gained widespread use in cosmetics, pharmacology, medicine, and the food industry. Mushrooms are recognized as rich natural sources of antioxidants. Accordingly, the antioxidant activity of Moroccan *F. fomentarius* was evaluated spectrophotometrically using three distinct assays, with results summarized in Table 4. The antioxidant capacity was expressed as EC_50_ (half maximal effective concentration) values, derived from dose–response curves presented in Appendix A. The methanolic extract of *F. fomentarius* exhibited potent antioxidant properties, with the strongest effect observed in the DPPH (2,2-diphenyl-1-picrylhydrazyl) radical-scavenging assay (EC_50_ = 114.40 µg/mL), followed by *β*-carotene bleaching inhibition (EC_50_ = 174.50 µg/mL) and ferric ion-reducing power (EC_50_ = 250.70 µg/mL) (Table 4). This robust antioxidant activity is attributed to the high content and diversity of Bioactive compounds identified in the Moroccan mushroom, primarily phenolic compounds (including phenolic acids, flavonoids, tannins), ascorbic acid, carotenoids, and, to a lesser extent, terpenoids, organic acids, fatty acids, and other volatile and non-volatile molecules. The antioxidant mechanism of these compounds, especially phenolics, involves electron donation or reducing activity, inhibition of oxidase enzymes, and chelation of metal ions such as Fe and Cu, which catalyze the production of reactive oxygen species (ROS) [22,25].

The antioxidant activities of *F. fomentarius* from various countries have been evaluated using diverse methods, with DPPH and ferric ion-reducing power assays being the most commonly employed. In contrast, the *β*-carotene bleaching inhibition assay has rarely been used in previous studies.

For the DPPH assay, the radical-scavenging activity of Moroccan *F. fomentarius* increased with concentration from 25 to 800 µg/mL, reaching a maximum inhibition of 90.58% at 800 µg/mL (Appendix A). This value slightly exceeds those reported for Turkish samples, which showed maximum scavenging rates of 80.93%, 88.88%, and 89.95% at 1 mg/mL for methanol, acetone, and ethanol extracts, respectively [43]. The EC_50_ value obtained here (114.40 µg/mL) indicates stronger antioxidant activity compared to prior reports of methanol extracts from Turkey (EC_50_ = 199.93 µg/mL) [17] and ethanol extracts from South Korea (EC_50_ = 500 µg/mL) [12]. Similarly, the Moroccan extract outperformed samples from Germany [44], Poland [24], Belarus [45], and Turkey [33], which had IC_50_ values ranging between 460 and 1390 µg/mL across different solvents such as methanol, ethanol, and ethyl acetate. Conversely, Serbian *F. fomentarius* extracts exhibited greater DPPH scavenging capacity, with EC_50_ values as low as 5.86 µg/mL reported by Kolundžić et al. [13] and 10.69 µg/mL (methanol) and 10.87 µg/mL (ethanol) by Karaman et al. [16]. Likewise, the ethanolic extract from Tunisia showed a notable IC_50_ of 26 µg/mL [18].

Regarding the *β*-carotene bleaching inhibition assay, the antioxidant activity of *F. fomentarius* increased with concentration up to 500 µg/mL, then plateaued between 70.11% and 85.88% from 500 to 2000 µg/mL (Appendix A). The EC_50_ of 174.50 µg/mL was considerably higher than that of Trolox (3.84 µg/mL), a synthetic antioxidant (Table 4).

In the ferric ion-reducing power assay, the methanolic extract demonstrated a steady increase in reducing ability, reaching an absorbance of 1.53 at 800 µg/mL (Appendix A). The EC_50_ was determined at 250.70 µg/mL, which is higher than the standard reference (80.11 µg/mL). The Moroccan extract showed greater reducing power than the ethanolic extract from Germany [44] and the methanolic extract from Turkey [40], which showed 40% and 2.09% activity at 320 µg/mL and 10,000 µg/mL, respectively. However, the Tunisian ethanolic extract displayed superior antioxidant capacity with an IC_50_ of 20 µg/mL [18]. Furthermore, Karaman et al. reported ferric ion-reducing power values of 136.60 mg/g and 78.76 mg/g ascorbic acid equivalents for methanol and ethanol extracts of Serbian *F. fomentarius*, respectively [16].

Pearson’s correlation coefficients (r) between antioxidant activities of the methanolic extract and bioactive compound contents, including nine major individual phenolic compounds, are shown in Figure 3. The ferric ion-reducing power assay (RPA) and DPPH scavenging activity were strongly and significantly correlated (*p* ≤ 0.05) with isorhamnetin, yielding r^2^ values of 1.000 and 0.999, respectively. Correlations between RPA and DPPH and kaempferol and TPC were high but statistically insignificant (*p* > 0.05), with r values ranging from 0.887 to 0.988. In contrast, RPA and DPPH showed negative correlations with LC, TFC, AAC, and βCC. For the *β*-carotene bleaching inhibition assay (IβCB), positive correlations were observed with TFC (0.970), LC (0.936), and AAC (0.798), while negative correlations were found with isorhamnetin (0.969), kaempferol (0.936), and TPC (0.779). Additionally, antioxidant activities exhibited weak correlations with TTC and p-hydroxybenzoic acid. These findings confirm the relationship between biomolecule compositions, especially polyphenols, and antioxidant activity [22,25].

### 2.5. Antimicrobial Properties of F. fomentarius

Following the analysis of bioactive compounds and antioxidant evaluation, the antimicrobial properties of the Moroccan *F. fomentarius* methanolic extract were assessed against seven human pathogens, comprising two bacterial and five fungal strains. The broth microdilution method, as recommended by the Clinical and Laboratory Standards Institute (CLSI), was employed to determine the minimum inhibitory concentration (MIC) and minimum bactericidal/fungicidal concentration (MBC/MFC). The results, summarized in Table 5 and Table 6, show that the methanolic extract exhibited considerable antimicrobial activity against all tested pathogens, with MIC values ranging from 2 to 32 mg/mL; however, its potency was lower than that of the synthetic antimicrobial agents, consistent with previous reports of the antimicrobial properties of *F. fomentarius* [13,16,18,19,40,43,45].

The antibacterial activity was approximately twofold higher against the Gram-positive *S. aureus* compared to the Gram-negative *E. coli*, with MIC values of 4 mg/mL and 8 mg/mL, and MBC values of 8 mg/mL and 16 mg/mL, respectively (Table 5).

The antifungal activity of the methanolic extract was evaluated against five human pathogenic fungi, including one yeast (*Candida albicans*), one filamentous fungus (*Aspergillus fumigatus*), and three dermatophytes. The extract inhibited all fungal pathogens, with MIC values ranging from 2 to 32 mg/mL (Table 6). Dermatophytes were more sensitive to the extract than *C. albicans* and *A. fumigatus*. Specifically, the extract showed a MIC of 32 mg/mL and MFC of 53.33 mg/mL against *C. albicans*, while *A. fumigatus* was the most resistant, with a MFC ≥64 mg/mL. Among dermatophytes, the strongest activity was against *Epidermophyton floccosum*, with MIC and MFC values of 2 mg/mL and 4 mg/mL, respectively. *Microsporum canis* was the most resistant dermatophyte (MIC = 4 mg/mL; MFC = 8 mg/mL), and *Trichophyton rubrum* showed MIC of 2 mg/mL and MFC of 7 mg/mL. Synthetic antifungal agents demonstrated significantly higher anti-dermatophyte activity than the mushroom extract, with terbinafine generally more potent than voriconazole.

Figure 4 presents a correlogram summarizing Pearson’s correlation coefficients between antimicrobial activities (MIC, MBC/MFC) of *F. fomentarius* methanolic extract and its main bioactive compounds. Generally, antimicrobial activities showed both positive and negative correlations with bioactive compound content, with some assays lacking significant correlations. Significant correlations (*p* ≤ 0.05) were observed between antibacterial activity against *E. coli* and *S. aureus* and the individual polyphenols protocatechuic acid, chlorogenic acid, catechin, and apigenin. These bacteria showed insignificant correlations with TFC, TTC, p-hydroxybenzoic acid, rutin, and quercetin, while correlations with TPC, AAC, and βCC were weak.

For antifungal activity, MIC and MFC values against *C. albicans* were significantly correlated with AAC (r = 0.997) and TPC (r = 0.999), respectively. Activity against *A. fumigatus* showed a significant negative correlation with TPC for MIC (−0.999, *p* ≤ 0.05) and a positive, albeit insignificant, correlation for MFC (0.829, *p* > 0.05). Among dermatophytes, strong correlations (r^2^ > 0.5) were found between extract activity against *T. rubrum* and most bioactive compounds, except for TFC, which had weak correlations. Significant and strong positive correlations were observed between antifungal activity against *E. floccosum* and *p*-hydroxybenzoic acid (r = 1.000) and TTC (r = 0.999) for MIC values, and chlorogenic acid (r = 1.000) and catechin (r = 0.997) for MFC values. Activity against *M. canis* showed generally high correlation coefficients with the nine major phenolic compounds. These results further confirm the strong relationship between bioactive compounds, mainly phenolic ones, and antimicrobial activity [22,23,24,25].

Antimicrobial activities of both edible and inedible mushrooms have been widely studied using various solvents, methods, and units of measurement, consistently demonstrating potent antimicrobial effects against diverse pathogens [46]. Similarly, *F. fomentarius* has been previously investigated for its antimicrobial potential, particularly antibacterial effects, yielding significant results [13,16,18,19,40,43,45].

Regarding antibacterial activity against *S. aureus* and *E. coli*, our results demonstrated higher efficacy than the ethanolic extract from Tunisian *F. fomentarius* (MIC = 12.5 mg/mL; MBC = 25 mg/mL) [18], yet lower activity compared to the Polish ethanolic extract (MIC and MBC between 1.25 and 5 mg/mL) [19]. Serbian *F. fomentarius* showed remarkable antibacterial activity against both bacteria using various solvents, with MIC and MBC ranging from 0.125 to 9 mg/mL [13,16]. Samples from Turkey [40] and Belarus [45] exhibited antibacterial activity against *S. aureus*, but not against *E. coli*.

Antifungal studies on *F. fomentarius* are scarce, but the following should be mentioned: *C. albicans* [45] and against *A. fumigatus* [47]. Notably, no prior investigations have assessed the antifungal activity of *F. fomentarius* against dermatophytes, the agents of ringworm. Contrary to our findings against *C. albicans*, a Belarusian study reported no activity of aqueous, chloroform, ethanol, or ethyl acetate extracts against this yeast [45]. Slama et al. found that aqueous extracts from Tunisian *F. fomentarius* were more inhibitory against *A. fumigatus* than methanolic extracts, with inhibition ranges of 61.11–64.83% and 23.08–34.62%, respectively [47]. Regarding dermatophytes, no prior data are available for comparison; however, the extract in this study was generally more effective than the methanolic extract of *Lactarius sanguifluus* from our previous research [48].

Overall, the notable antimicrobial effects observed in this study can be attributed to the diverse biomolecules present in *F. fomentarius*, including phenolic compounds, terpenoids (mainly steroids), organic acids, and other volatile and non-volatile molecules. The observed pharmacological activities are likely influenced not only by the major compounds, which have been previously reported to exhibit biological activity, but also by minor constituents that may contribute through synergistic interactions between different chemical classes [49]. These compounds likely disrupt microbial growth by inhibiting cell wall synthesis, nucleic acid production, and protein synthesis [25]. In summary, the volatile and non-volatile biomolecules in the methanolic extract of *F. fomentarius* exhibited potent antimicrobial activity against seven tested human pathogens. These findings suggest that *F. fomentarius* is a promising source of antimicrobial agents for treating various microbial infections.

## 3. Materials and Methods

### 3.1. Chemical Reagents and Standards

Phenolic compounds standards (≥95%), linoleic acid (≥98%), gentamicin (≥99%), Trolox ((±)-6-Hydroxy-2,5,7,8-tetramethylchromane-2-carboxylic acid) (97%), *β*-carotene (≥95%), l-ascorbic acid (99%), meta-phosphoric acid (≥33.5% (T)), MOPS (≥99.5%), sodium carbonate (≥99.5%), alkane standards (C_8_–C_20_ and C_21_–C_40_) (99.5%), *Folin–Ciocalteu* phenol reagent (>99%), Tween 40 (~90.0%), sodium nitrite (≥99%), BSTFA (N,O-Bis(trimethylsilyl)trifluoroacetamide) (≥99%), 2,6-Dichloroindophenol sodium salt hydrate (≥95%), iron (III) chloride (≥99.99%), sodium hydroxide (≥98%), dimethyl sulfoxide (DMSO) (≥99.9%) and vanillin reagent (99%) were purchased from Sigma-Aldrich, Co., (St. Louis, MO, USA). The media Mueller–Hinton Broth (MHB), Mueller–Hinton Agar (MHA), and Sabouraud Dextrose Agar (SDA) were obtained from BioMérieux (Marcy L’Étoile, France), while Potato Dextrose Agar (PDA) from Difco Laboratories (Detroit, MI, USA), and RPMI-1640 medium was from Biochrom AG (Berlin, Germany). 2,2-diphenyl-l-picrylhydrazyl (DPPH) (95%) was bought from Alfa Aesar (Ward Hill, MA, USA), and acetonitrile (≥99.9%), hydrochloric acid fuming 37%, aluminum chloride (≥99.99%), pyridine (≥99.9%), ethyl acetate (≥99.7%), and sodium chloride (≥99%) were acquired from Merck KGaA (Darmstadt, Germany). Terbinafine (≥98%) was obtained from Novartis Pharma AG (Basel, BS, Switzerland), and voriconazole (99.91%) from Pfizer Inc. (New York, NY, USA). Methanol and all other solvents (≥99.9%) were purchased from Honeywell (St. Muskegon, MI, USA) and CABLO ERBA Reagent, S.A.S. (Val de Reuil Cedex, France).

### 3.2. Mushroom Material

The tinder fungus, *Fomes fomentarius* (L.: Fr.) Kickx was collected in 2018 on *Acacia* species, specifically *A. rubida* and *A. saligna*, within the Koudiat Taifour Forest, a designated Biological and Ecological Interest Site (SIBE) located in northern Morocco (35°40′45.4″ N, 5°17′36.3″ W, at an altitude of 180 m). This site is characterized by a siliceous shale substrate, a thermo-Mediterranean vegetation zone, and a subhumid bioclimatic level with temperate winters. The specimen was identified based on ecological and morphological characteristics using two taxonomic determination keys [50,51]. After identification, the voucher specimen was deposited in the herbarium of the BEDD laboratory at the ENS, Abdelmalek Essaâdi University, Tetouan, Morocco. For further investigation, *F. fomentarius* was cultivated on potato dextrose agar (PDA) medium at 25 °C and preserved for future studies. The collected fruiting bodies were sliced, air-dried, and ground into a fine powder. Extraction was performed using methanol following the same protocol described in our previous study [4]. The resulting extracts were dried, weighed, and stored at −81 °C for further use. The extraction yield (%) was calculated using the following equation: Yield (%) = [Extract weight after solvent evaporation/Dry weight of fruiting body] × 100.

### 3.3. Determination of Bioactive Compound Contents

The determination of bioactive compound contents, including TPC, TFC, TTC, AAC, and βCLC, was performed spectrophotometrically using the same equipment, conditions, and methods previously described by Erbiai et al. [52].

The *Folin–Ciocalteu* assay was employed to estimate the TPC in the methanolic extract of *F. fomentarius*. A sample solution (1 mL, 2 mg/mL) was added to tubes containing 5 mL of *Folin–Ciocalteu* reagent (diluted 1:10, *v*/*v*) and 4 mL of sodium carbonate solution (75 g/L). The mixture was thoroughly homogenized and incubated at 40 °C for 30 min. Absorbance was then measured at 765 nm using methanol as a blank. Gallic acid was used as the reference standard, and the obtained value was expressed as mg GAE/g of dme.

The aluminum chloride (AlCl_3_) colorimetric method was used to quantify the TFC in *F. fomentarius* by mixing a volume of methanolic extract solution (0.5 mL, 2.5 mg/mL) with 0.15 mL of sodium nitrite (NaNO_2_) solution (5%) and 2 mL of distilled water, then allowing it to stand for 6 min. Subsequently, 0.15 mL of AlCl_3_ solution (10%) was added, and after another 6 min, 2 mL of sodium hydroxide (NaOH) solution (4%) and 0.2 mL of distilled water were introduced. The obtained mixture was thoroughly homogenized and left to stand for 15 min. The absorbance of the developed color was measured at 510 nm against a blank. (+)-Catechin was used as the reference standard to construct the calibration curve, and the TFC was expressed as mg CE/g of dme.

The vanillin–HCl method was followed to determine the TTC in *F. fomentarius*. One gram of the powdered sample was extracted with 50 mL of methanol at 28 °C for 24h, followed by centrifugation at 3000 × g for 10 min. From the resulting supernatant, 1 mL was mixed with 5 mL of hydrochloric reagent, consisting of 1% vanillin and 8% concentrated HCl prepared in methanol (1:1, *v*/*v*). The mixture was allowed to react for 20 min, and absorbance was measured at 500 nm. The TTC was expressed as mg CE/g of dme.

The AAC in Moroccan *F. fomentarius* was estimated using a method based on the reaction between ascorbic acid and 2,6-dichlorophenolindophenol (2,6-DCIP). One gram of mushroom powder was extracted with 10 mL of 1% (*w*/*v*) meta-phosphoric acid for 45 min at room temperature. The extract was then filtered, and 100 µL of the filtrate was mixed with 900 µL of 2,6-DCIP solution. Absorbance was measured at 515 nm within 30 min. The vitamin C content was expressed as mg AAE/g dw.

The quantification of carotenoids, namely *β*-carotene and lycopene, was performed by re-extracting 100 mg of dried methanolic extract in 10 mL of an acetone–hexane mixture (4:6, *v*/*v*) for 1 min. The extract was then filtered, and the absorbance (A) of the filtrate was measured simultaneously at four wavelengths: 453 nm, 505 nm, 645 nm, and 663 nm. *β*CLC were calculated using the following formulas: *β*-carotene (mg/100 mL) = [(0.216 A_663_) − (0.304 A_505_) + (0.452 A_453_)]; Lycopene (mg/100 mL) = [(0.0458 A_663_) + (0.372 A_505_) − (0.0806 A_453_)].

### 3.4. Biomolecules Analysis of F. fomentarius Methanolic Extract by GC-MS

The bioactive compound profile of the methanolic extract of the medicinal mushroom *F. fomentarius* was determined using GC–MS. Prior to analysis, the methanolic extract was prepared as three different samples: (1) derivatized with BSTFA reagent in chloroform [53], and (2) and (3) diluted directly in methanol and chloroform, respectively. The GC–MS analysis was performed using a TRACE 1300 GC system (Thermo Fisher Scientific, Waltham, MA, USA) coupled with an ISQ single quadrupole MS detector (ISQ Single Quadrupole GC-MS; Thermo Fisher Scientific, Waltham, MA, USA) equipped with an automated injector. GC separation was carried out on a TG5-MS capillary column (60 m × 0.25 mm i.d., 0.25 μm film thickness; 5% phenyl/95% dimethylpolysiloxane) (Thermo Fisher Scientific, Waltham, MA, USA). The injector and detector temperatures were set at 300 °C, using splitless injection mode (1:10). Helium was used as a carrier gas at a flow rate of 0.75 mL/min. The oven temperature was programmed from 50 °C (2 min) to 200 °C at 5 °C/min (2 min hold), then ramped at 5 °C/min to 350 °C (6 min hold). The total run time was 65 min. Mass spectrometric conditions: electron ionization (EI) at 70 eV was used in full scan acquisition mode over an *m*/*z* range of 50–650. The ion source temperature was set at 300 °C, the quadrupole temperature at 250 °C, and the transfer line temperature at 280 °C. The biomolecules identification was conducted based on the Kovats retention indices (RI), calculated using a homologous series of standard alkanes (C_8_–C_20_ and C_21_–C_40_). Further structural confirmation was achieved by comparing spectral data with reference libraries, including the National Institute of Standards and Technology (NIST) database and online repositories such as NIST and PubChem. Data acquisition and analysis were carried out using Thermo Xcalibur™ 2.2 SP1.48 (Thermo Fisher Scientific, Waltham, MA, USA) and NIST MS Search 2.2 Library (2014) (NIST, Gaithersburg, MD, USA). The results were expressed as the relative percentage of each detected biomolecule [53].

### 3.5. Characterization of Individual Polyphenols by LC-MS Analysis

The extraction of polyphenolic compounds from *F. fomentarius* was performed by mixing 1 g of finely powdered mushroom material with 20 mL of methanol:water (80:20, *v*/*v*). The mixture was incubated at −20 °C for 2 h, followed by sonication for 15 min, centrifugation at 4000× *g* for 10 min, and filtration through Whatman No. 4 filter paper. The residue was re-extracted using the same procedure to maximize yield. The combined methanolic extracts were concentrated by evaporating methanol, leaving an aqueous phase, which was then subjected to liquid–liquid extraction using diethyl ether (2 × 20 mL) and ethyl acetate (2 × 20 mL). The remaining water was removed by adding anhydrous sodium sulfate. Afterward, the organic phase was evaporated to dryness at 40 °C, and the extracts were reconstituted in methanol:water (80:20, *v*/*v*), followed by filtration through a 0.22 µm disposable LC filter disk to prepare the sample for high-performance liquid chromatography (HPLC) analysis.

Concerning the chromatographic analysis, individual polyphenol compounds were identified and quantified following previously published work by our research team [4], using the same HPLC system (Thermo Scientific, Waltham, MA, USA) and analytical conditions. Briefly, the polyphenol extract was analyzed by liquid chromatography-mass spectrometry (LC-MS). Chromatographic separation was performed on an Acclaim™ 120 reverse-phase C18 column (3 µm, 150 × 4.6 mm) (Thermo Fisher Scientific, Waltham, MA, USA) maintained at 35 °C. The detection wavelength for peak identification was set at 280 nm. The mobile phase consisted of 1% acetic acid (solvent A) and 100% acetonitrile (solvent B). Polyphenol detection was carried out using a photodiode array (PDA) detector at 280 nm, and individual compounds were identified by chromatographic comparison with commercial authentic standards. Quantification was performed using calibration curves generated from authentic standards at six concentrations (5–160 µg/mL), all of which showed high linearity (R^2^ > 0.995). Limits of detection (LOD) and limits of quantification (LOQ), defined as signal-to-noise ratios of 3 and 10, respectively, ranged from 0.05 to 0.16 µg/mL (LOD) and 0.05 to 0.25 µg/mL (LOQ). Recoveries ranged from 90% to 95%. Procedural and solvent blanks were analyzed under the same conditions, and pooled quality control samples were injected every ten runs to monitor instrument stability. All measurements were performed in triplicate, and the results were expressed as µg/g dry weight (dw).

### 3.6. Determination of Antioxidant Properties

The antioxidant properties of the bioactive compound constituents in the methanolic extract of *F. fomentarius* were evaluated using three spectrophotometric assays: DPPH radical-scavenging activity, *β*-carotene bleaching inhibition, and the reducing power assay. Trolox was used as the reference standard in all tests. Antioxidant activity was expressed as EC_50_ values (half-maximal effective concentration), calculated from dose–response curves, representing the extract or standard concentration required to achieve 50% antioxidant activity or 0.5 absorbance [54].

#### 3.6.1. DPPH Radical-Scavenging Activity (RSA)

RSA of the methanolic extract of *F. fomentarius* was evaluated using the stable DPPH^•^ (2,2-diphenyl-1-picrylhydrazyl) free radical assay. A 0.3 mL aliquot of the extract at a known concentration was mixed with 2.7 mL of DPPH solution in methanol in a test tube. The mixture was then incubated in the dark at room temperature for 30 min, after which the absorbance was measured at 517 nm. Methanol was used as the blank, while the DPPH solution served as the control. The RSA percentage was calculated using the following equation: RSA (%) = [(A*_DPPH_*− A*_ES_*)/A*_DPPH_*] × 100, where A*_ES_* is the absorbance of the extract sample, and A*_DPPH_* is the absorbance of the control DPPH solution.

#### 3.6.2. Inhibition of *β*-Carotene Bleaching (I*β*CB)

The *β*-carotene-linoleate model system was employed as the second method to assess the antioxidant activity of biomolecules present in the methanolic extract of *F. fomentarius*. A 4.8 mL aliquot of freshly prepared *β*-carotene emulsion was mixed with 0.2 mL of the extract at different concentrations. The mixture was thoroughly stirred, and the initial absorbance (A_0_) was measured at 470 nm against a blank containing the emulsion without *β*-carotene. The test tubes were then covered and incubated in a water bath at 50 °C with shaking at 100 rpm for 2 h. After incubation, the absorbance (A_2_h) was measured again at 470 nm. A control sample was prepared under the same conditions by replacing the extract with 0.2 mL of methanol. Extract blanks (extract without *β*-carotene) were also prepared and their absorbance was subtracted from the corresponding test values. The *β*-carotene bleaching inhibition (I*β*CB, %) was calculated using the following equation: I*β*CB (%) = (*β*-carotene content after 2 h of assay)/(initial *β*-carotene content) × 100.

#### 3.6.3. Reducing Power Assay (RPA)

The RPA was employed as the third method to evaluate the antioxidant capacity of the methanolic extract of *F. fomentarius*, based on its ability to reduce Fe^3+^ to Fe^2+^ using the ferricyanide/Prussian blue method. For the assay, 1.5 mL of the extract at a known concentration was mixed with 1.5 mL of phosphate buffer (0.2 M, pH 6.6) and 1.5 mL of 1% potassium ferricyanide solution. The mixture was then incubated at 50 °C for 20 min, followed by the addition of 1.5 mL of 10% trichloroacetic acid (TCA). The resulting solution was centrifuged at 1000 rpm for 8 min, and 1.5 mL of the upper layer was carefully transferred to a new tube. Afterward, 1.5 mL of distilled water and 0.3 mL of 0.1% ferric chloride solution were added to the supernatant. Finally, the absorbance was measured at 690 nm against a blank. Higher absorbance values indicate greater reducing power of the extract.

### 3.7. Evaluation of Antimicrobial Properties

The antimicrobial properties of the biomolecules present in the methanolic extract of *F. fomentarius* fruiting bodies were assessed using the broth microdilution method, following the CLSI guidelines (M07-A8 for bacteria, M27-A3 for yeasts, and M38-A2 for filamentous fungi) [55,56,57,58]. This method enables the determination of MIC and subsequently the MBC or MFC.

The experimental procedures, reagents, standards, conditions, equipment, and pathogenic strains used in this study were identical to those described in our previous work [4]. The antimicrobial activity was evaluated against the following human pathogens: *E. coli* ATCC 25922 and *S. aureus* ATCC (bacteria), *A. fumigatus* ATCC 46645 (filamentous fungus), *C. albicans* ATCC 10231 (yeast), and *E. floccosum* FF9, *M. canis* FF1, and *T. rubrum* FF5 (Dermatophytes). For quality control, *C. krusei* ATCC 6258 was included. All pathogenic strains used in this study were obtained from the Microbiology Laboratory, Faculty of Pharmacy, University of Porto (Portugal).

Concerning the determination of MIC and MFC/MBC, the dried methanolic extract and synthetic antimicrobial agents were dissolved in dimethyl sulfoxide (DMSO) at a final concentration of ≤2%, followed by serial dilutions in the appropriate growth media: RPMI-1640 with MOPS (pH 7.0) for fungi and Mueller-Hinton broth (MHB) for bacteria to achieve the necessary concentration range. A 100 µL aliquot of each diluted sample was transferred to the wells of a 96-well microplate, followed by the addition of 100 µL of microbial suspension (1–2 × 10^5^ colony-forming units (CFU)/mL for bacteria, 1–5 × 10^3^ CFU/mL for yeast, 0.4–5 × 10^4^ CFU/mL for *Aspergillus* and 1–3 × 10^3^ CFU/mL for dermatophytes), that had been prepared by dilution in fresh MHB for bacteria and RPMI-1640 for fungi. Microplates were then incubated without agitation at 37 °C for 24 h for the two bacteria, 48 h for *Candida* and *Aspergillus*, and seven days at 25 °C for the dermatophyte isolates. The MIC was defined as the lowest concentration at which no visible microbial growth was observed. To determine MBC/MFC, 10 µL from wells showing no turbidity in the MIC assay was subcultured onto MHA for bacteria and SDA for fungi. The MBC/MFC was recorded as the lowest concentration that completely inhibited microbial growth under the same incubation conditions. The synthetic antimicrobial agents, namely gentamicin, voriconazole, and terbinafine, were employed as positive controls.

### 3.8. Statistical Analysis

Except for GC-MS analysis, all assays were conducted in triplicate using three independent samples, and the results were expressed as mean ± standard deviation (SD). The correlation between bioactive compound content, major phenolic compounds, and pharmacological properties was assessed using Pearson’s correlation coefficient (r). Statistical analyses were performed using GraphPad Prism 8.0.1 (San Diego, CA, USA).

## 4. Conclusions

This study represents the first investigation of the biomolecule composition and pharmacological properties of *F. fomentarius* from Morocco, revealing it as a rich source of biologically active compounds with diverse biological activities. The fruiting body extracts contained notable bioactive compounds, including phenolic compounds, ascorbic acid, sugars, fatty acids, terpenoids, organic acids, alcohols, and various other volatile and non-volatile molecules. These compounds exhibited potent antioxidant activity and significantly inhibited the growth of all seven tested human pathogenic microorganisms. Collectively, these findings suggest that Moroccan *F. fomentarius* could serve as a valuable source of bioactive agents with potential applications in human health and well-being.

## Figures and Tables

**Figure 1 ijms-26-09215-f001:**
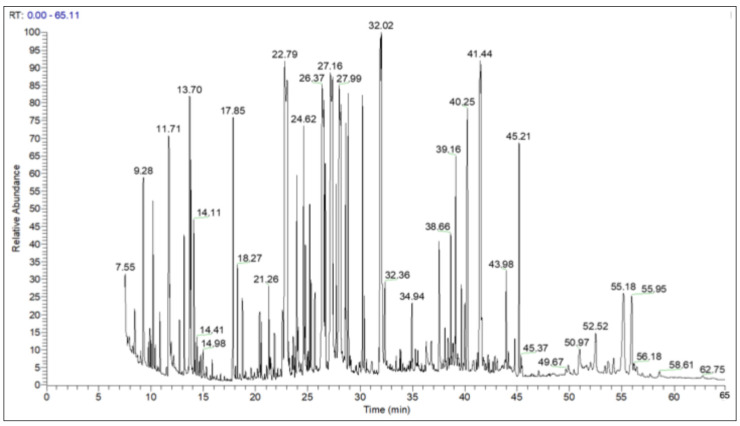
GC–MS chromatogram of the biomolecules profile of *F. fomentarius* derivatized methanolic extract.

**Figure 2 ijms-26-09215-f002:**
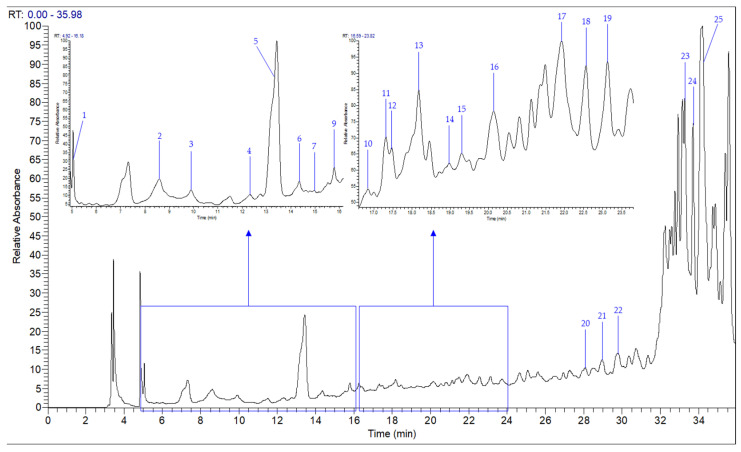
LC–MS chromatogram at 280 nm of individual polyphenols in hydromethanolic extract of *F. fomentarius*.

**Figure 3 ijms-26-09215-f003:**
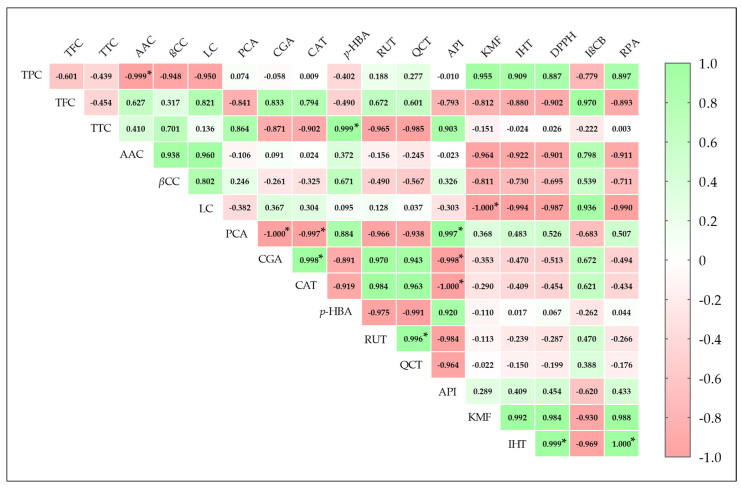
Correlogram illustrating PCC (Pearson’s correlation coefficient (r)) between main bioactive compounds (TPC, TFC, TTC, AAC, βCC, LC, and nine major individual polyphenols) and antioxidant activity (EC_50_ values of DPPH, IβCB, and RPA). Positive correlations are presented in green and negative ones in red. PCC values were mentioned in colored squares, and the color intensity is directly proportional to the correlation coefficient. * Indicate significant correlations between variables (*p* ≤ 0.05). Total phenolic (TPC), total flavonoid (TFC), total tannin (TTC), ascorbic acid (AAC), *β*-carotene (*β*CC), lycopene (LC), protocatechuic acid (PCA), chlorogenic acid (CGA), catechin (CAT), *p*-hydroxybenzoic acid (*p*-HBA), rutin (RUT), quercetin (QCT), apigenin (API), kaempferol (KMF), isorhamnetin (IHT), DPPH radical-scavenging activity (DPPH), *β*-carotene bleaching inhibition (IβCB), and Ferric ion-reducing power (RPA).

**Figure 4 ijms-26-09215-f004:**
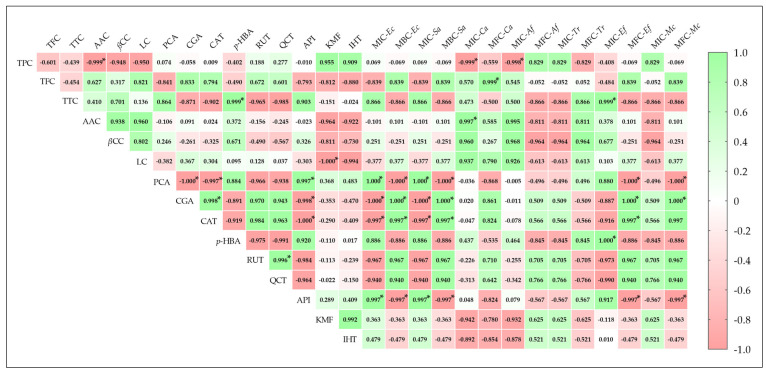
Correlogram representing Pearson’s correlation coefficient between main bioactive compounds (TPC, TFC, TTC, AAC, βCC, LC, and nine major individual polyphenols) and antimicrobial activity (values of minimum inhibitory concentration (MIC) and minimum bactericidal/fungicidal concentration (MBC/MFC)). Positive correlations are presented in green, and negative ones in red. PCC values were mentioned in colored squares, and the color intensity is directly proportional to the correlation coefficient. * Indicate significant correlations between variables (*p* ≤ 0.05). *E. coli* (MIC-Ec & MBC-Ec), *S. aureus* (MIC-Sa & MBC-Sa), *C. albicans* (MIC-Ca & MFC-Ca), *A. fumigatus* (MIC-*Af* & MFC-*Af*), *T. rubrum* (MIC-*Tr* & MFC-*Tr*), *E. floccosum* (MIC-*Ef* & MFC-*Ef*), and *M. canis* (MIC-*Mc* & MFC-*Mc*).

**Table 1 ijms-26-09215-t001:** Extraction yield and bioactive compound contents of methanol extract and phenolic compounds in the dried fruiting body of *F. fomentarius*.

Bioactive Compounds Content	*F. fomentarius*
Extraction yield (%)	7.59 ± 0.15 ^1^
Total phenolic content (mg GAE/g dme)	75.83 ± 0.33
Total flavonoid content (mg CE/g dme)	37.62 ± 1.09
Total tannin content (mg CE/g dw)	0.95 ± 0.07
Ascorbic acid content (mg AAE/g dw)	3.43 ± 0.07
*β*-carotene content (mg/g dme)	0.59 ± 0.00
Lycopene content (mg/g dme)	0.19 ± 0.00

^1^ Values are expressed as mean ± standard deviation (SD) of three independent measurements (*n* = 3).

**Table 2 ijms-26-09215-t002:** Groups of biomolecules identified in the three samples of *F. fomentarius* methanolic extracts by GC–MS analysis.

Groups of Biomolecules	Dilution in Chloroform		Derivatized with BSTFA	
	Area (%)	Number of Molecules	Area (%)	Number of Molecules
Alcohols	1.06	5	3.27	5
Alkaloids	11.55	2	-	-
Alkanes	1.66	7	-	-
Amino acids	-	-	1.08	6
Fatty acids	15.80	12	14.72	11
Organic acids	3.09	2	5.36	7
Phenols	40.30	3	0.14	1
Steroids	0.31	1	3.75	3
Sugar compositions	-	-	64.63	24
Other groups	32.99	7	7.03	13
Total	98.65	39	99.98	70

**Table 3 ijms-26-09215-t003:** Quantified phenolic compounds of *F. fomentarius* hydromethanolic extract analyzed by LC–MS in negative mode ^1^.

Peak No.	Rt (min)	MW	Recorded *m*/*z*	Individual Polyphenols	Content (µg/g dw)
1	5.05	170.02	169.67	Gallic acid	25.42 ± 0.06
2	8.60	154.12	153.10	Protocatechuic acid	98.61 ± 0.61
3	9.90	354.31	352.78	Chlorogenic acid	150.80 ± 0.86
4	12.31	290.08	289.51	Catechin	116.90 ± 2.43
5	13.42	138.03	137.45	*p*-Hydroxybenzoic acid	409.00 ± 2.46
6	14.35	180.04	179.01	Caffeic acid	4.96 ± 0.06
7	14.96	168.04	167.86	Vanillic acid	2.19 ± 0.06
8	-	-	-	Syringic acid	ND
9	15.78	610.15	608.78	Rutin	95.27 ± 0.25
10	16.84	302.20	303.01	Ellagic acid	5.56 ± 0.15
11	17.31	448.37	448.32	Luteolin 7-glucoside	21.85 ± 0.16
12	17.47	164.05	164.97	*p*-Coumaric acid	5.78 ± 0.05
13	18.17	152.05	151.07	Vanillin	15.81 ± 0.03
14	18.98	194.19	193.17	Ferulic acid	3.34 ± 0.02
15	19.30	580.18	578.18	Naringin	2.51 ± 0.03
16	20.14	432.11	432.61	Apigenin 7-glucoside	26.26 ± 0.14
17	21.92	360.08	361.03	Rosmarinic acid	17.15 ± 0.55
18	22.56	138.03	137.03	Salicylic acid	31.90 ± 0.13
19	23.13	152.05	151.94	Methyl paraben	21.46 ± 0.13
20	28.08	286.23	285.12	Luteolin	45.34 ± 0.34
21	28.96	302.04	301.35	Quercetin	99.47 ± 1.09
22	29.78	148.05	148.14	Cinnamic acid	11.02 ± 0.31
23	33.27	270.05	269.15	Apigenin	295.10 ± 0.24
24	33.71	286.05	285.25	Kaempferol	351.10 ± 0.56
25	34.19	316.27	315.26	Isorhamnetin	2734.00 ± 7.33

^1^ Results are expressed as mean ± SD of three independent measurements. Rt—retention time; dw—dried weight; ND—undetected.

**Table 4 ijms-26-09215-t004:** EC_50_ values (µg/mL) of antioxidant activities of the methanolic extracts of *F. fomentarius* fruiting body ^1^.

Antioxidant Assay	*F. fomentarius*	Trolox
DPPH radical-scavenging activity	114.40 ± 2.57 ^Ca^	19.17 ± 0.99 ^Bb^
*β*-carotene bleaching inhibition	174.50 ± 1.17 ^Ba^	3.84 ± 0.70 ^Cb^
Ferric ion-reducing power	250.70 ± 1.75 ^Aa^	80.11 ± 2.37 ^Ab^

^1^ EC_50_ values are shown as mean ± SD (*n* = 3). Different superscript uppercase and lowercase letters indicate statistically significant differences within rows and columns, respectively (*p* < 0.05).

**Table 5 ijms-26-09215-t005:** MIC and MBC values of *F. fomentarius* methanolic extract and standard reference against bacterial strains ^1^.

Bacteria Strains	*F. fomentarius* Extract (mg/mL)		Gentamicin (mg/mL)	
	MIC	MBC	MIC	MBC
*Escherichia coli*	8.00 ± 0.00 ^Ab^	16.00 ± 0.00 ^Ab^	0.0020 ± 0.00 ^Ac^	>0.032
*Staphylococcus aureus*	4.00 ± 0.00 ^Bb^	8.00 ± 0.00 ^Ba^	0.0004 ± 0.00 ^Bd^	0.016 ± 0.00 ^c^

^1^ Values expressed as mean ± SD (*n* = 4–6). MIC—minimum inhibitory concentration; MBC—minimum bactericidal concentration. For each bacterium and sample (MIC and MBC), different superscript uppercase and lowercase letters indicate statistically significant differences within rows and columns, respectively (*p* < 0.05).

**Table 6 ijms-26-09215-t006:** MIC and MFC values of *F. fomentarius* methanolic extract and reference antifungal agents ^1^.

Fungal Pathogen Strains	*F. fomentarius* Extract (mg/mL)		Voriconazole (mg/mL)		Terbinafine (mg/mL)	
	MIC	MFC	MIC	MFC	MIC	MFC
*Candida albicans*	32.00 ± 0.00 ^Ab^	53.33 ± 18.48 ^Aa^	0.00038 ± 0.0 ^Ac^	>0.004	-	-
*Aspergillus fumigatus*	26.67 ± 9.24 ^Ba^	≥64	0.00025 ± 0.0 ^Bc^	0.00075 ± 0.29 ^Bb^	-	-
*Trichophyton rubrum*	2.00 ± 0.00 ^Db^	7.00 ± 1.14 ^Ba^	0.00013 ± 0.0 ^Cd^	0.0015 ± 0.58 ^Ac^	0.00001 ± 0.0 ^Ce^	0.000129 ± 0.0 ^Ce^
*Epidermophyton floccosum*	2.00 ± 0.00 ^Db^	4.00 ± 0.00 ^Ca^	0.00003 ± 0.0 ^De^	0.00009 ± 0.04 ^Dd^	0.00002 ± 0.0 ^Bf^	0.000038 ± 0.0 ^Be^
*Microsporum canis*	4.00 ± 0.00 ^Cb^	8.00 ± 0.00 ^Ba^	0.00013 ± 0.0 ^Ce^	0.00038 ± 0.14 ^Cd^	0.00012 ± 0.0 ^Ae^	0.00075 ± 0.0 ^Ac^

^1^ Values expressed as mean ± SD (*n* = 4–6). MIC—minimum inhibitory concentration; MFC—minimum fungicidal concentration. For each fungus and sample (MIC and MBC), different superscript uppercase and lowercase letters indicate statistically significant differences within rows and columns, respectively (*p* < 0.05).

## Data Availability

Data are contained within the article or Appendix A.

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
