# Peer review of "Bioactive Compounds and Pharmacological Properties of the Polypore *Fomes fomentarius*, a Medicinal Wild Mushroom Collected from Morocco"

_ijms, 2025, doi:10.3390/ijms26189215_

Round 1
Reviewer 1 Report
Comments and Suggestions for Authors
Dear Authors,
Please find my recommendations listed below for "Mycochemicals and Pharmacological Properties of the Polypore Fomes fomentarius, a Medicinal Wild Mushroom Collected from Morocco" manuscript.
- L45: I suggest for authors to replace "mycochemicals" with other standard terminology as "secondary metabolites" or "bioactive compounds." Please consider this recommendation through the whole text of the manuscript
- L47: Please reconsider this "Polypore mushrooms are a diverse group of edible or non-toxic fungi" as this classification represents imprecise mycological terminology in my opinion. Many polypores are neither edible nor toxic but simply inedible due to ligneous texture and lack of nutritional value
- L67-69: Here the authors oversimplified the complex parasitic-saprophytic transition in my opinion. Please avoid that
- The introduction analysis is insufficient to clarify the gap identified in the research. Please consider rephrasing it and conducting a deeper literature review
- In the current format the "Results and discussions" seems to be more "Results" section; In my opinion the interpretation of the obtained results should be well discussed and interpreted. Please provide mechanistical insight for the obtained results (explain them, interpret them) - for example please better discuss the structure-activity relationships and mechanistic explanations for observed bioactivities; better ensure their contextualization within existing literature; better consider the synergistic effects between compounds; and better improve the correlation analysis between chemical composition and biological activity. Also ensure comparison with related studies from literature (provide more critical comparisons)
- I recommend to avoid such enumerative presentation "[13,16,18,19,40,42,44]" L404 of the literature citing, instead please integrate them in discussions
- L432: As the authors decided to list the used chemicals/standards used please provide also the corresponding purity
- Please provide information about quality controls for instrumental analysis. For individual polyphenol compounds LC-MS analysis please include methods performance data also (LOD, LOQ, recovery, etc.)
- L505: Please include the mass spectrometric analysis parameters from GC-MS analysis - as ionization mode, detector and transfer line temperature, mass spectra acquisition mode full scan/SIM), etc.
- Please consider these recommendations and provide enough details/technical details for the next presented methods also
- After considering the improvements, please reconsider the conclusion section
Author Response
Response to Reviewer 1 Comments
Thank you very much for taking the time to review this manuscript. Please find the detailed responses below and the corresponding revisions/corrections highlighted in the re-submitted files.
Point-by-point response to Comments and Suggestions for Authors
Dear Authors,
Please find my recommendations listed below for "Mycochemicals and Pharmacological Properties of the Polypore Fomes fomentarius, a Medicinal Wild Mushroom Collected from Morocco" manuscript.
Comments 1: L45: I suggest for authors to replace "mycochemicals" with other standard terminology as "secondary metabolites" or "bioactive compounds." Please consider this recommendation through the whole text of the manuscript
Response 1: We appreciate your observations and suggestions. We have replaced it accordingly.
Comments 2: L47: Please reconsider this "Polypore mushrooms are a diverse group of edible or non-toxic fungi" as this classification represents imprecise mycological terminology in my opinion. Many polypores are neither edible nor toxic but simply inedible due to ligneous texture and lack of nutritional value
Response 2: We appreciate your suggestion and helpful comment. We have revised it accordingly.
Comments 3: L67-69: Here the authors oversimplified the complex parasitic-saprophytic transition in my opinion. Please avoid that
Response 3: We appreciate your observation and suggestion. We have revised it accordingly.
Comments 4: The introduction analysis is insufficient to clarify the gap identified in the research. Please consider rephrasing it and conducting a deeper literature review
Response 4: We appreciate your comment and suggestion. We have revised it accordingly.
Comments 5: In the current format the "Results and discussions" seems to be more "Results" section; In my opinion the interpretation of the obtained results should be well discussed and interpreted. Please provide mechanistical insight for the obtained results (explain them, interpret them) - for example please better discuss the structure-activity relationships and mechanistic explanations for observed bioactivities; better ensure their contextualization within existing literature; better consider the synergistic effects between compounds; and better improve the correlation analysis between chemical composition and biological activity. Also ensure comparison with related studies from literature (provide more critical comparisons)
Response 5: We appreciate your comment and suggestion. We have revised it accordingly.
Comments 6: I recommend to avoid such enumerative presentation "[13,16,18,19,40,42,44]" L404 of the literature citing, instead please integrate them in discussions
Response 6: We appreciate your comment and recommendation. This sentence was intended as an introduction to the discussion and was meant to be general. After that, the cited literature can be found integrated separately throughout the discussion.
Comments 7: L432: As the authors decided to list the used chemicals/standards used please provide also the corresponding purity.
Response 7: We appreciate your comment and suggestion. We have revised it accordingly.
Comments 8: Please provide information about quality controls for instrumental analysis. For individual polyphenol compounds LC-MS analysis please include methods performance data also (LOD, LOQ, recovery, etc.)
Response 8: We appreciate this important comment. We have revised it accordingly.
Comments 9: L505: Please include the mass spectrometric analysis parameters from GC-MS analysis - as ionization mode, detector and transfer line temperature, mass spectra acquisition mode full scan/SIM), etc.
Response 9: We appreciate this useful suggestion. We have revised accordingly.
Comments 10: Please consider these recommendations and provide enough details/technical details for the next presented methods also.
Response 10: We appreciate your recommendations. We have revised accordingly
Comments 11: After considering the improvements, please reconsider the conclusion section
Response 11: We appreciate your comments and suggestions. We have revised accordingly
Reviewer 2 Report
Comments and Suggestions for Authors
In this manuscript, the author identified and quantified the bioactive substances in F. fomentarius, and evaluated the antioxidant and antibacterial properties of F. fomentarius methanol extract. Overall, this manuscript is not yet in a publishable state in International Journal of Molecular Sciences. It needs major revisions.
- Strain identification relies solely on morphological characteristics, lacking molecular biological confirmation. To ensure accurate species authentication, it is essential to supplement this with sequencing.
- The extracts were not subjected to a decolorization procedure in the antioxidant assays. This is a particular concern for the β-carotene bleaching inhibition assay, where intrinsic pigments in the extract could interfere with the absorbance measurements at 470 nm, potentially leading to an overestimation of antioxidant activity.
- The rationale for selecting methanol as the extraction solvent is not sufficiently justified. The authors should briefly explain the basis for this choice to demonstrate its
- In the GC-MS analysis, results from derivatized and underivatized samples are simplistically combined and reported as a total of "109 compounds." These two sample preparation methods target distinct groups of compounds with different volatilities and polarities. The results should be clearly differentiated and discussed separately.
- In Table 5, the units for the fomentariusextract are mg/mL, while those for the Gentamicin standard are μg/mL. The authors' statement in the results section “The results, summarized in Tables 5 and 6, show that the methanolic extract exhibited considerable antimicrobial activity against all tested pathogens”, The units should be unified to allow for a clear and accurate comparison of potency. Furthermore, references should be supplemented to support "considerable antimicrobial activity".
- The observed pharmacological activities are the combined effect of a complex mixture of hundreds of compounds. Attributing the activity simply to the major compound classes (e.g., phenolics) is an oversimplification. The contributions of minor constituents or potential synergistic effects between components are significant possibilities that are not addressed.
- Units are mixed in the manuscript, such as "hours"(line 486)and "h"(line 612).
- The manuscript lacks a dedicated section discussing the limitations of the study.
Author Response
Response to Reviewer 2 Comments
Thank you very much for taking the time to review this manuscript. Please find the detailed responses below and the corresponding revisions/corrections highlighted in the re-submitted files.
In this manuscript, the author identified and quantified the bioactive substances in F. fomentarius, and evaluated the antioxidant and antibacterial properties of F. fomentarius methanol extract. Overall, this manuscript is not yet in a publishable state in International Journal of Molecular Sciences. It needs major revisions.
Comments 1: Strain identification relies solely on morphological characteristics, lacking molecular biological confirmation. To ensure accurate species authentication, it is essential to supplement this with sequencing.
Response 1: We appreciate your valuable comment and observation. We fully agree that molecular tools provide definitive confirmation of species identity. In our study, the identification of Fomes fomentarius was based on its well-documented macroscopic, microscopic, and ecological characteristics, which are widely recognized as reliable diagnostic features for this species. The specimens were also verified by expert mycologists (co-authors) in our laboratory. Unfortunately, due to the unavailability of molecular biology facilities and resources during this study, we were unable to perform molecular identification.
Comments 2: The extracts were not subjected to a decolorization procedure in the antioxidant assays. This is a particular concern for the β-carotene bleaching inhibition assay, where intrinsic pigments in the extract could interfere with the absorbance measurements at 470 nm, potentially leading to an overestimation of antioxidant activity.
Response 2: We appreciate your important observation. We agree that colored compounds present in the F. fomentarius extract could contribute to the absorbance at 470 nm and influence the β-carotene bleaching results. Although a decolorization step was not applied, we minimized this effect by including extract blanks (extract without β-carotene) prepared under the same conditions, and their absorbance was subtracted from the test samples to correct for any intrinsic color interference. We have added this point in the revised method.
Comments 3: The rationale for selecting methanol as the extraction solvent is not sufficiently justified. The authors should briefly explain the basis for this choice to demonstrate its
Response 3: We appreciate your observation and comment. Methanol was selected as the extraction solvent primarily because it was readily available in our laboratory, and more importantly, because it is widely used for extracting phenolic compounds and other polar bioactive metabolites from fungal matrices. Methanol efficiently penetrates fungal cell walls, and its high polarity and excellent solubility for a broad range of bioactive molecules make it particularly suitable for this type of analysis. This approach has also been well documented in several previous studies on fungal extracts.
Comments 4: In the GC-MS analysis, results from derivatized and underivatized samples are simplistically combined and reported as a total of "109 compounds." These two sample preparation methods target distinct groups of compounds with different volatilities and polarities. The results should be clearly differentiated and discussed separately.
Response 4: We appreciate this insightful comment and observation. We agree that the derivatized and underivatized samples represent distinct chemical fractions with different volatilities and polarities. In our study, these two sample types were analyzed, interpreted, and discussed separately; the total number of identified biomolecules was mentioned only to indicate the overall diversity detected in the mushroom extracts. As shown in the manuscript, each sample is presented separately (39 compounds in the underivatized extract and 70 in the derivatized extract), along with their major chemical groups and representative compounds (Table 2 and Supplementary Tables S1–S13). If the reviewer considers it clearer, we are willing to remove the combined total count to avoid any potential confusion.
Comments 5: In Table 5, the units for the fomentarius extract are mg/mL, while those for the Gentamicin standard are μg/mL. The authors' statement in the results section “The results, summarized in Tables 5 and 6, show that the methanolic extract exhibited considerable antimicrobial activity against all tested pathogens”, The units should be unified to allow for a clear and accurate comparison of potency. Furthermore, references should be supplemented to support "considerable antimicrobial activity".
Response 5: We appreciate your observation and comment. We have revised and corrected accordingly.
Comments 6: The observed pharmacological activities are the combined effect of a complex mixture of hundreds of compounds. Attributing the activity simply to the major compound classes (e.g., phenolics) is an oversimplification. The contributions of minor constituents or potential synergistic effects between components are significant possibilities that are not addressed.
Response 6: We appreciate your observation and this valuable comment. We agree with your observation and comment, and we have revised it accordingly.
Comments 7: Units are mixed in the manuscript, such as "hours"(line 486)and "h"(line 612).
Response 7: We appreciate your observation and comment. We have revised it accordingly (using “h” for hours).
Comments 8: The manuscript lacks a dedicated section discussing the limitations of the study.
Response 8: We appreciate your observation and comment. We have revised it accordingly
Reviewer 3 Report
Comments and Suggestions for Authors
The manuscript is well-written, clearly structured, and presents a comprehensive study of Fomes fomentarius. The experimental design and methodologies are appropriate and rigorously applied, combining detailed chemical characterization with relevant pharmacological assays. The results are clearly presented and interpreted, providing strong evidence for the antioxidant and antimicrobial potential of the species. Overall, the manuscript is scientifically sound and makes a valuable contribution to the field.
Table 4, 5 and table 6: Statistical data in tables is missing. Please add statistical significant differences - since You stated that all tests were conducted in triplicates.
Line 234: This is not true, there is also one study where bioactive potential and mychochemical composition of F. fomentarius Balkan strains was assesed - please refer to this study as well:Rašeta, M.; Kebert, M.; Pintać Šarac, D.; Mišković, J.; Berežni, S.; Kulmány, Á.E.; Zupkó, I.; Karaman, M.; Jovanović-Šanta, S. Bioactive Potential of Balkan Fomes fomentarius Strains: Novel Insights into Comparative Mycochemical Composition and Antioxidant, Anti-Acetylcholinesterase, and Antiproliferative Activities. Microorganisms 2025, 13, 1210. https://doi.org/10.3390/microorganisms13061210
Author Response
Response to Reviewer 3 Comments
Thank you very much for taking the time to review this manuscript. Please find the detailed responses below and the corresponding revisions/corrections highlighted in the re-submitted files.
The manuscript is well-written, clearly structured, and presents a comprehensive study of Fomes fomentarius. The experimental design and methodologies are appropriate and rigorously applied, combining detailed chemical characterization with relevant pharmacological assays. The results are clearly presented and interpreted, providing strong evidence for the antioxidant and antimicrobial potential of the species. Overall, the manuscript is scientifically sound and makes a valuable contribution to the field.
Comments 1: Table 4, 5 and table 6: Statistical data in tables is missing. Please add statistical significant differences - since You stated that all tests were conducted in triplicates.
Response 1: We appreciate your observation and comment. We have revised and added accordingly
Comments 2: Line 234: This is not true, there is also one study where bioactive potential and mychochemical composition of F. fomentarius Balkan strains was assesed - please refer to this study as well: Rašeta, M.; Kebert, M.; Pintać Šarac, D.; Mišković, J.; Berežni, S.; Kulmány, Á.E.; Zupkó, I.; Karaman, M.; Jovanović-Šanta, S. Bioactive Potential of Balkan Fomes fomentarius Strains: Novel Insights into Comparative Mycochemical Composition and Antioxidant, Anti-Acetylcholinesterase, and Antiproliferative Activities. Microorganisms 2025, 13, 1210. https://doi.org/10.3390/microorganisms13061210
Response 2: We thank the reviewer for pointing this out. We have revised the manuscript and included the study from the Balkan region (Croatia, Serbia, and Bosnia and Herzegovina).
Round 2
Reviewer 2 Report
Comments and Suggestions for Authors
The authors have responded satisfactorily to the comments provided.